# Critic-Driven Decoding for Mitigating Hallucinations in Data-to-text Generation

**Mateusz Lango** and **Ondřej Dušek**
Charles University, Faculty of Mathematics and Physics
Institute of Formal and Applied Linguistics
Prague, Czech Republic
{lango,odusek}@ufal.mff.cuni.cz

## Abstract

Hallucination of text ungrounded in the input is a well-known problem in neural data-to-text generation. Many methods have been proposed to mitigate it, but they typically require altering model architecture or collecting additional data, and thus cannot be easily applied to an existing model. In this paper, we explore a new way to mitigate hallucinations by combining the probabilistic output of a generator language model (LM) with the output of a special "text critic" classifier, which guides the generation by assessing the match between the input data and the text generated so far. Our method does not need any changes to the underlying LM's architecture or training procedure and can thus be combined with any model and decoding operating on word probabilities. The critic does not need any additional training data, using the base LM's training data and synthetic negative examples. Our experimental results show that our method improves over the baseline on the WebNLG and OpenDialKG benchmarks.

## 1 Introduction

Hallucination, i.e., generated text lacking grounding in the input data, is a major challenge in neural data-to-text generation (Raunak et al., 2021; Rebuffel et al., 2022; Corbelle et al., 2022; Ji et al., 2023). Hallucinations can lead to inaccurate or misleading information, significantly undermining the quality and reliability of the generated output. While many approaches have been proposed to address this problem, they often involve modifying the underlying model architecture (Rebuffel et al., 2022) or acquiring additional data (Wang, 2019; Thomson et al., 2020), making them impractical for existing models. At the same time, popular metrics for evaluating hallucinations are based on text classification models, e.g. NLI-based metrics (Honovich et al., 2021; Dušek and Kasner, 2020). This indicates that text classifiers have the potential to accurately identify and assess coherence problems between the data and the generated text. However, use of text classifiers in generation typically involves producing many outputs with a base model and reranking them afterwards (Harkous et al., 2020).

In this paper, we propose a novel *critic-driven decoding* approach that combines the probabilistic output of a conditional language model (LM) with the output of a specialized *text critic* classifier that guides the generation process by evaluating the coherence of the textual prefix generated so far with the input data. This allows us to influence the generation on-the-fly, without the need to overgenerate many outputs. Furthermore, our approach does not require modifications to the underlying LM or additional fine-tuning. This ensures compatibility with a wide range of existing models and decoding algorithms that operate on word probabilities. Finally, our method does not rely on the collection of additional data, as training data for the critic can be synthesized from the data-to-text dataset used to train the underlying conditional LM.

We verify the effectiveness of our critic-driven decoding in experiments on the WebNLG (Gardent et al., 2017) and OpenDialKG (Moon et al., 2019) benchmarks, with both automatic and manual evaluation of text hallucinations in the model outputs. The results show that our method is able to limit hallucinations and produce a more faithful text, yet close to the base LM's output. Our implementation of the proposed method is publicly available.[1]

## 2 Critic-driven decoding

Recall that auto-regressive conditional LMs for data-to-text generation rely on the following probability factorization:

$$P(y|x) = \prod_{i=1}^{n} P(y_i|y_{\leq i-1}, x) \qquad (1)$$

---

[1] https://github.com/langus0/critic-aware-decoding

where $x$ is the data representation and $y$ is the generated text. We use $y_{\leq j}$ to denote all the tokens $y_1, y_2, y_3, ..., y_j$.

In our approach, we introduce to this probability an additional text generation critic which evaluates the match between the generated text and the data representation. The output of the critic $c$ can be seen as a binary variable, equal to 1 if the text matches the input data and 0 otherwise. This leads to the following probability factorization:

$$P(y|x, c) = \prod_{i=1}^{n} P(y_i|y_{\leq i-1}, x, c) \qquad (2)$$

i.e. generation of text $y$ given the data representation $x$ and the output of the critic $c$. By applying simple probability transformations (see Appendix A), we obtain the following factorization:

$$P(y_i|y_{\leq i-1}, x, c) \propto P(c|y_{\leq i}, x)P(y_i|y_{\leq i-1}, x) \qquad (3)$$

This formulation combines two probabilities: the probability of a standard conditional LM $P(y_i|y_{\leq i-1}, x)$ and the probability of the match between the text and the data as evaluated by the critic model $P(c|y_{\leq i}, x)$.

The critic is modelled as a binary classifier, conditioned on the data, past tokens and the token currently being decoded. It is thus run at each decoding step, and it is evaluating the viability of the current prefix of the generation output (assuming future generation will be successful). The proposed formulation can be used with any auto-regressive conditional LM without modification, as the operation is identical to Eq. 1. The critic can be trained separately from the LM since our formulation implies the conditional independence of those two models.

The above factorization leads us to a practical proposal of a critic-driven decoding algorithm. First, an additional critic model is trained, which is able to approximate $P(c|y_{\leq i}, x)$ (details are discussed in Sec. 3). We then perform standard greedy decoding with the LM, but using the updated formula for calculating probabilities of the next tokens (Eq. 3). In practice, our implementation operates on logarithms rather than raw probabilities and uses an additional weight $\lambda$ that adjusts the influence of the critic on the final result:

$$\ln P(y_i|y_{\leq i-1}, x, c)$$
$$\propto \lambda \ln P(c|y_{\leq i}, x) + \ln P(y_i|y_{\leq i-1}, x) \qquad (4)$$

# 3 Training a text generation critic

The critic model $P(c|y_{\leq i}, x)$ is a binary classifier that checks the correspondence between the linearized data representation $x$ and the so far generated prefix of the output text $y_{\leq i}$. We assume an encoder pretrained LM as the backbone of the critic. The model input contains $x$ and $y_{\leq i}$ split by a separator token.

Positive instances for the critic's training are constructed from examples $(x, y)$ in the underlying LM's dataset as prefixes: $(x, y_1), (x, y_{\leq 2}), (x, y_{\leq 3}), ..., (x, y_{\leq n})$. Negative examples must be synthesized and are crucial for training the critic, as they teach it how to detect that the generated text starts deviating from the input data (i.e. hallucinations). Therefore, we explore five ways of generating negative examples (see Appendix G for examples):

1. *base* – for each positive example, we replace the last token with a random one. To make the training set more challenging, the tokens are sampled from another text reference for the same data (if available) or another random sentence from the dataset.

2. *base with full sentences* – a randomly selected sentence of the reference text $y$ is replaced with a random sentence from the dataset. Negative examples are then generated in the same way as positive examples, but starting from the first token that deviates from the reference. In addition, instances where a random token in the reference is replaced by a wrong one are also generated in the same way.

3. *vanilla LM* – for each positive example we probe an unconditional LM to get a list of the five most likely next tokens. We randomly select a token from this list and construct a negative example.

4. *fine-tuned LM* – similar to the previous, but using the LM conditioned on the data.

5. *fine-tuned LM with full sentences* – the LM conditioned on the data is used to generate a textual description of the data. The negative examples are constructed for each token starting from the one where the model starts deviating from the reference.

All critic model variants are trained by optimizing binary cross-entropy loss.

## 4 Experimental evaluation

We compare all critic variants defined in Sec. 3 with the baseline LM on both automatic and manual metrics, focusing on the number of hallucinations.

### 4.1 Experimental setup

We performed most experiments on the WebNLG benchmark (Gardent et al., 2017) containing data expressed as RDF triples and corresponding text references, which is prominent in many previous works tackling hallucination. We also evaluate our approach on the OpenDialKG dataset (Moon et al., 2019), which contains dialogues annotated with RDF triples representing the information expressed in each utterance. We use it in a similar way as WebNLG, treating the utterances as textualisations of the data, i.e. without taking dialogue history into account. The BART-base encoder-decoder model (Lewis et al., 2020), finetuned on WebNLG, is used as the base NLG system (see Appendix C for training details).

Five different critic models were trained as discussed in Sec. 3 with classification heads on top of a XLM-RoBERTa-base model (Conneau et al., 2019), see Appendix D for details. The vanilla LM critic uses BART-base without any fine-tuning, the fine-tuned LM variants (4 & 5) use the base LM to generate training data. The critics' classification performance is given in Table 1. This shows that the critics are able to learn the synthetic data well, which is, however, not necessarily reflected in performance when used during generation.

We use greedy decoding by default. To speed up computation of critic-driven decoding, we first evaluate the second term of Eq. 3, i.e. the conditional LM, and we run the critic model only for $k = 5$ most probable tokens, modifying its probabilities accordingly. The critic weight $\lambda = 0.25$ (see Eq. 4) was used for all the models for WebNLG and $\lambda = 1$ for OpenDialKG. We found that the output of the critic can be noisy when evaluating the match between the data and only a few initial tokens of the text. Therefore, we add a simple linear warmup for $\lambda$ for the first five tokens: while decoding the $i$-th token, $\lambda_i = \min(\frac{i}{5}, 1) \cdot \lambda$ (cf. Appendix B for choice of $k$ and warmup).

### 4.2 Analysis of decoding performance with automatic measures

The system outputs were evaluated using standard automatic metrics – BLEU (Papineni et al.,

| critic model | accuracy | F1 |
|---|---|---|
| 1. base | 0.969 | 0.970 |
| 2. base w/full sent. | 0.984 | 0.975 |
| 3. vanilla. LM | 0.931 | 0.798 |
| 4. fine-tuned LM | 0.920 | 0.718 |
| 5. fine-tuned LM w/full sent. | 0.929 | 0.714 |

Table 1: The classification performance of different critic models as measured on the validation test.

2002), METEOR (Banerjee and Lavie, 2005) and BERTScore (Zhang et al., 2020) – as well as measures particularly targeting hallucinations: BLEURT (Sellam et al., 2020) and the NLI-based metric proposed by Dušek and Kasner (2020).

**Overall results on WebNLG** are presented in Table 2. Except for the critic trained on full LM-generated sentences (var. 5), all the other variants of critic-driven decoding slightly improve performance according to BLEU, METEOR, and BERTScore. Higher gains, up to 2.5% absolute on the whole test set, can be observed on measures targeting hallucinations, i.e. NLI and BLEURT. Note that our approach achieves this without modifying the original LM. The base critic achieves the highest scores across most evaluation metrics.

Interestingly, both critics trained on data generated with the fine-tuned LM (i.e. the same system as used for decoding) failed to improve the NLI measure and only one improved BLEURT. This shows that an effective critic can be trained separately from the NLG system.

**Analysis of introduced changes** We also measured to what extent the critic-based approaches change the outputs compared to the baseline, i.e. the percentage of modified outputs as well as the number of added and removed words.[2] Results in Tab. 4 show that critic-based approaches preserve many outputs (30-70%) and generally only change a few words, while keeping the output lengths similar. This suggests that our critics make small changes and only where necessary.

**Out of domain generalization** The test data of the WebNLG dataset contains about 46% of instances from categories not present in the training data. Therefore, we also provide the fine-grained results for both in-domain and out-domain part of the test set in Table 2. The in-domain results are naturally better, but we can observe consistent im-

---

[2]Replacing a word counts as one addition and one deletion.

| decoding approach | BLEU | METEOR | BERT Score | NLI | | | BLEURT | | |
|---|---|---|---|---|---|---|---|---|---|
| | | | | all | ood | ind | all | ood | ind |
| baseline | 45.09 | 0.373 | 0.911 | 0.841 | 0.783 | 0.889 | 0.128 | -0.026 | 0.257 |
| 1. critic (base) | 45.48 | **0.377** | **0.913** | 0.855 | 0.801 | 0.901 | **0.155\*** | **0.010\*** | **0.277\*** |
| 2. critic (base with full sentences) | 44.90 | 0.371 | **0.913** | **0.868\*** | **0.820\*** | **0.909** | 0.153* | 0.007* | 0.274 |
| 3. critic (vanilla LM) | 45.44 | **0.377** | **0.913** | 0.859* | 0.811 | 0.900 | 0.139 | -0.002 | 0.258 |
| 4. critic (fine-tuned LM) | 45.41 | 0.373 | 0.911 | 0.834 | 0.772 | 0.886 | 0.128 | -0.021 | 0.254 |
| 5. critic (fine-tuned LM w. full sentences) | **45.59** | 0.374 | 0.912 | 0.839 | 0.779 | 0.889 | 0.136 | -0.013 | 0.261 |

Table 2: Results of automatic evaluation on the WebNLG test set. NLI and BLEURT are reported for the whole test set (*all*) as well as separately for its out-of-domain (*ood*) and in-domain (*ind*) parts. "\*" marks statistical significance at $\alpha = 0.05$ level (NLI: exact test for proportions, BLEURT: unpaired t-test).

| | BLEU | METEOR | BERTScore | NLI | BLEURT |
|---|---|---|---|---|---|
| baseline | 11.74 | 0.149 | 0.775 | 0.748 | -0.933 |
| 1. critic (base) | 9.67 | 0.137 | 0.771 | **0.796** | **-0.905** |
| 2. critic (base with full sentences) | **11.88** | **0.151** | **0.776** | 0.754 | -0.920 |
| 3. critic (vanilla LM) | 10.37 | 0.139 | 0.763 | 0.713 | -0.980 |
| 4. critic (fine-tuned LM) | 10.76 | 0.143 | 0.768 | 0.739 | -0.964 |
| 5. critic (fine-tuned LM with full sentences) | 11.41 | 0.149 | 0.771 | 0.712 | -0.956 |

Table 3: Results of automatic evaluation on the OpenDialKG test set.

provements of our critic-aware approach on both in-domain and out-of-domain data.

**Statistical analysis** We performed statistical hypothesis testing to compare the results of the baseline with our approach with critic (base with full sentences). As expected, the differences on text quality measures (BLEU, METEOR, BERTScore) are not statistically significant, in contrast to the differences on measures targeting hallucinations, which are statistically significant at the $\alpha = 0.05$ significance level (cf. Table 2).

**Beam search experiment** To verify the consistency of our critic's improvements, we run additional experiments with a stronger baseline, i.e. beam search decoding. The results, confirming greedy decoding results, are in Appendix F.

**Results on OpenDialKG** are presented in Table 3 and are mostly in line with those obtained for WebNLG. The base critic approach (var. 1) obtained a gain of 5 percentage points on NLI and of 3 points on BLEURT over the baseline. The values of basic word-overlap metrics are lower, but our qualitative assessment did not confirm any quality drop. The second critic variant (base with full sentences), which offered high performance of WebNLG, also performed well on OpenDialKG. It scored best on standard text quality metrics while offering improvements over the baseline on hallucination-focused metrics (NLI, BLEURT).

| critic model | mod [%] | add. | rem. |
|---|---|---|---|
| base | 66.3 | 4.54 | 4.58 |
| base w/full sent. | 72.8 | 5.42 | 4.72 |
| vanilla LM | 72.8 | 5.03 | 5.39 |
| fine-tuned LM | 48.5 | 2.52 | 2.71 |
| fine-tuned LM w/full sent. | 31.9 | 1.63 | 1.76 |

Table 4: Percentage of modified outputs and average number of words added/removed by different critics compared to standard decoding on WebNLG.

### 4.3 Manual analysis of decoding performance

To verify the automatic metric results, we performed a small-scale in-house human evaluation. We sampled 100 instances from the test set of the WebNLG dataset and annotated for them the output of all the systems under study (600 system outputs in total). The annotation was performed by five NLP expert annotators, who assessed the presence of minor hallucinations (mostly typos in named entity names), major hallucinations (output containing fact(s) not supported by the data), omissions (missing information), disfluencies (grammar errors or hard-to-read text) and repetitions (information mentioned twice). Finally, the annotators ranked the system outputs for each example from best to worst, with ties allowed. The annotation was blind, with system order randomly shuffled for each example. Results are summarised in Table 5 (see Appendix E for inter-annotator agreement).

All critic-driven approaches achieved better av-

| decoding approach | min. hal. | maj. hal. | omi. | disfl. | rep. | avg. rank |
|---|---|---|---|---|---|---|
| baseline | 0.22 | 0.40 | 0.25 | 0.20 | 0.08 | 3.61 |
| 1. critic (base) | 0.21 | 0.30 | **0.20** | 0.17 | **0.04** | **3.38** |
| 2. critic (base with full sentences) | 0.21 | **0.29** | 0.27 | **0.11** | 0.08 | 3.43 |
| 3. critic (vanilla LM) | **0.18** | **0.29** | 0.23 | 0.19 | 0.05 | 3.54 |
| 4. critic (fine-tuned LM) | 0.22 | 0.37 | 0.26 | 0.21 | 0.07 | 3.53 |
| 5. critic (fine-tuned LM with full sentences) | 0.20 | 0.37 | 0.26 | 0.18 | 0.07 | 3.54 |

Table 5: Results of manual evaluation on a sample of 100 examples from the WebNLG test set (percentage of examples with minor and major hallucinations, omissions, disfluencies, repetitions; average relative ranking).

erage ranks than the baseline, with the base critic (var. 1) having the best rank. The rank difference compared to the baseline is not large (0.23), but increases for more complex instances: in instances with three or more triples, the difference is 0.33, for instances with file or more triples, it is 0.53. More importantly, the base critic reduced the rate of major hallucination by 10% absolute. Again, the improvements are bigger for more complex instances (15.3% for $\geq 3$ triples, 20% for $\geq 5$). It also performed better on all other criteria, producing a more fluent output, with fewer omissions and repetitions, as well as a slightly reduced number of minor hallucinations.

Other critic variants were also effective in reducing hallucinations; in particular, the vanilla LM critic (var. 3) was the most effective in reducing both major and minor hallucinations. The fine-tuned LM approaches (vars. 4 & 5) only provided very limited benefits.

## 5   Related works

Hallucination in NLG is a widely studied problem, with many different mitigation methods proposed, including data cleaning or various model architecture modifications (see Ji et al. (2023) for a detailed review). Mitigation methods most similar to ours include the controlled generation approach by Filippova (2020), which uses special control codes to control hallucinations. This was followed by Rashkin et al. (2021), who combine control codes with resampling of several texts and selecting the best one according to the metrics. However, both approaches require training a new LM with control codes and, in the latter case, additional resampling of whole texts. Cao et al. (2020) proposed a two-step generate & refine procedure, which is model-independent but requires training of an additional correcting LM and decoding the sequence twice. Similarly to our approach, Chen et al. (2021) use a text classifier to select the best output among the so-called contrast candidates but does not use it during decoding.

Our method is closely inspired by works on class-conditional LMs, which use the Bayes rule to introduce additional conditioning (Cohn-Gordon et al., 2018; Dathathri et al., 2020). In particular, a formulation similar to ours is used by FUDGE (Yang and Klein, 2021) to impose a certain requirement, such as a level of formality, on the text produced by a LM. However, these works do not address the issue of hallucinations.

The use of randomly generated words as negative samples to improve natural language generation has also been explored by Welleck et al. (2020). In contrast to this work, their unlikelihood training technique is mainly aimed at limiting repetitive text generation and requires training a new model, as it modifies the training objective.

## 6   Summary

Our paper introduces a novel critic-driven decoding approach to mitigate hallucinations in data-to-text generation. By using the output of a specialised text critic classifier, we guide the generation process to produce more grounded output without requiring any modifications to the underlying LM. The experimental results on the WebNLG and OpenDialKG benchmarks show that the proposed method has the potential to limit hallucinations without hindering other text quality metrics.

## Acknowledgments

This work was supported by the European Research Council (Grant agreement No. 101039303, NG-NLG) and used resources of the LINDAT/ CLARIAH-CZ Research Infrastructure (Czech Ministry of Education, Youth, and Sports project No. LM2018101). The authors would like to thank Ondřej Plátek, Patrícia Schmidtová and Sourabrata Mukherjee, who kindly provided manual annotations for this work.

## Limitations

While our approach strives to remove as many hallucinations as possible from the NLG output, a certain proportion still remains for all our setups. The performance of the approach is limited by the base LM and its proposed most likely next tokens (as a limited number of next tokens is considered at each step, cf. Sec. 4). Furthermore, the use of the critic slows down the decoding. For application to other datasets, the critic may become less effective if the underlying training data is too noisy.

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

## A Derivation of the proposed probability factorization that incorporates a critic model

By applying the conditional probability formula and the product rule to Eq. 2, we obtain the following:

$$
\begin{aligned}
P(y_i|y_{\leq i-1}, x, c) &= \\
&= \frac{P(y_i, y_{\leq i-1}, x, c)}{P(y_{\leq i-1}, x, c)} \\
&= \frac{P(c|y_i, y_{\leq i-1}, x)P(y_i|y_{\leq i-1}, x)P(y_{\leq i-1}, x)}{P(y_{\leq i-1}, x, c)} \\
&= P(c|y_{\leq i}, x)P(y_i|y_{\leq i-1}, x)\frac{P(y_{\leq i-1}, x)}{P(y_{\leq i-1}, x, c)} \\
&= P(c|y_{\leq i}, x)P(y_i|y_{\leq i-1}, x)P(c|y_{\leq i-1}, x)^{-1} \\
&\propto P(c|y_{\leq i}, x)P(y_i|y_{\leq i-1}, x)
\end{aligned}
$$

where the last line comes from the fact that when computing the probability of the next token $y_i$, the previous tokens $y_{\leq i-1}$ are fixed, so the critic's score for the previous tokens $P(c|y_{\leq i-1}, x)$ is a constant and does not affect the result.

## B Sensitivity analysis of the hyperparameters of critic-aware decoding

### B.1 The number of most probable considered tokens

To speed up computations of critic-driven decoding, we run the critic model only for $k$ most probable tokens according to the LM and modify its probabilities with Eq. 3. The results in the paper are reported for $k = 5$, but we performed additional experiments with $k = 15$ to investigate how it will affect the performance. The results are given in Table 6. In general, we observe minor differences in comparison to $k = 5$. Some metrics has been slightly improved, but it probably does not counterbalance the additional computational cost.

| decoding approach | BLEU | METEOR | BERTScore | NLI | BLEURT |
|---|---|---|---|---|---|
| baseline | 45.09 | 0.373 | 0.911 | 0.841 | 0.128 |
| 1. critic (base) | 45.57 | **0.378** | **0.914** | 0.857 | **0.157** |
| 2. critic (base with full sentences) | 44.96 | 0.371 | 0.913 | **0.867** | 0.155 |
| 3. critic (unconditional LM) | 45.53 | 0.377 | 0.913 | 0.865 | 0.141 |
| 4. critic (conditional LM) | 45.41 | 0.373 | 0.911 | 0.834 | 0.129 |
| 5. critic (conditional LM with teacher forcing) | **45.59** | 0.374 | 0.912 | 0.839 | 0.136 |

Table 6: Results of automatic evaluation on WebNLG dataset for $k = 15$

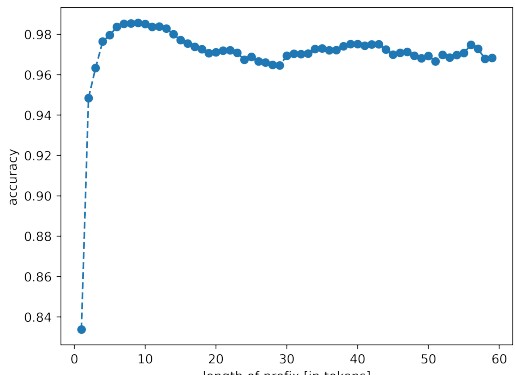

Figure 1: The accuracy of the critic (base) as a function of prefix length on validation set

## B.2 The importance of linear warm-up

A practical motivation for using linear warm-up of the $\lambda$ parameter can be found in Figure 1, which shows the accuracy as a function of text prefix length for one of the critic models (base, var. 1). It can be observed that at the beginning of the generation process (i.e. for very short prefixes) the accuracy of the critic is low, but grows rapidly with the length of the prefix, reaching a high level around the length of 5 tokens.

The importance of linear warm-up is investigated by comparing the decoding performance with a constant $\lambda$ and with linear warm-up (i.e. $\lambda_i = \min(\frac{i}{5}, 1) \cdot \lambda$). The results of this experiment for BLEU and BLEURT measures are depicted in Figure 2 and 3, respectively. It can be observed that the linear warm-up provides better performance for almost every model.

## C Hyperparameters of BART fine-tuning

As a conditional language model, we used BART-base model (Lewis et al., 2020) fine-tuned with default architecture provided by HuggingFace library. AdamW with learning rate $\eta = 2 \cdot 10^{-5}$ and parameters $\beta = (0.9, 0.997)$, $\epsilon = 10^{-9}$ was used

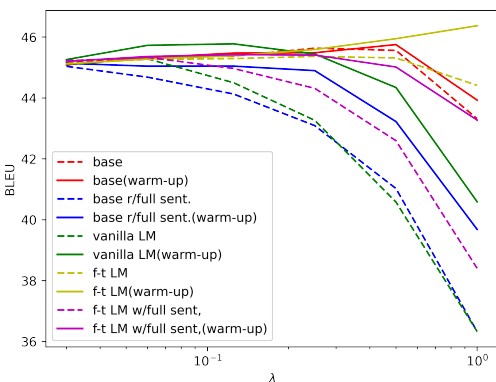

Figure 2: BLEU as a function of $\lambda$ parameter for system outputs generated with different critic variants and with/without warm-up of $\lambda$.

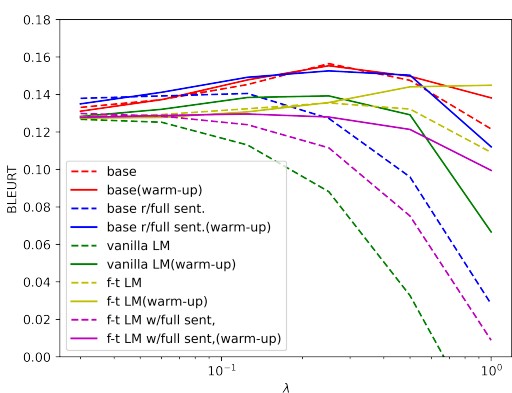

Figure 3: BLEURT as a function of $\lambda$ parameter for system outputs generated with different critic variants and with/without warm-up of $\lambda$.

as optimizer. Additionally, we applied polynomial scheduler of $\eta$ with a warmup equal to 10% of optimization steps. The training was scheduled for 20 epochs with early stopping on validation loss (patience of 10 epochs). We used batch size equal to 8 and label smoothing with 0.1 smoothing factor.

## D   Details on critic model training

The architecture of the critic model consisted of a pretrained XLM-RoBERTa-base model (Conneau et al., 2019) and a classification head on top of the representation of the first token. The classification head contained a fully connected layer with SELU activation function (Klambauer et al., 2017) and one additional classification layer with sigmoid activation. The number of neurons in the first layer was set to the dimensionality of the output embedding.

The critic models were trained as discussed in Sec. 3 by optimizing the cross-entropy loss. AdamW was used as an optimizer with a learning rate of $\eta = 10^{-5}$. The training was continued until the convergence, i.e. lack of the improvement on validation loss.

All the experiments with the critics (both critic training and decoding) were performed on one GPU: nVidia Quadro P5000 16 GB. During decoding the BART-based language model was loaded with bitsandbytes library (8-bit mode).

## E   Inter-annotator agreement

To estimate the inter-annotator agreement, one of the annotators re-annotated 10 ($\times$ 6 model outputs) instances originally annotated by a different annotator. 86% of annotations were identical. In terms of Cohen's kappa, 0.19 agreement was obtained for minor hallucinations, 0.68 for major, 0.88 for omissions, 0.48 for repetitions and 0.07 for disfluencies.

## F   Comparison with a stronger baseline

One simple method which generates multiple outputs and generally tends to offer texts of higher quality is beam search. We run additional experiments with beam size equal to 5 and present the result in the Table 7. The improvements for this stronger baseline are consistent with these reported in the main paper for greedy decoding.

## G   Examples of negative instances generated by different approaches for critic training set construction

Let us consider the following data representation:

(A-Rosa Luna | length | 125800.0 (millimetres));
(A-Rosa Luna | power type | MTU Friedrichshafen)

and the reference:

The A-Rosa Luna is powered by a MTU Friedrichshafen engine and is 125.8 metres in length.

The positive examples for the critic consist on all the possible prefixes generated from the reference, i.e. "The", "The A-Rosa", "The A-Rosa Luna", etc. The negative examples generated by different approaches are as follows:

1. base – the negative examples are generated with random words

   "The Cruises", "The A-Rosa operated", "The A-Rosa Luna located", ...

2. base with full sentences - a sentence or a token from the reference is replaced with random sentence/token and all possible negative examples are generated

   "The Cruises", "The Cruises Luna", "The Cruises Luna is", ..., "The A-Rosa operated", "The A-Rosa operated is", ...

3. vanilla LM – the incorrect next words are sampled from the five most probable tokens according to (unconditioned) LM

   "The United", "The A-Rosa is", "The A-Rosa Luna powers", ...

4. fine-tuned LM with full sentences – for a given data the NLG system generated the following output: "The A-Rosa Luna is 125.8m long and is powered by MTU Friedrichsburger", which is used to generate negative examples by comparing it against the reference

   "The A-Rosa Luna is 125.8m", "The A-Rosa Luna is 125.8m long", "The A-Rosa Luna is 125.8m and", "The A-Rosa Luna is 125.8m and is", ...

| | BLEU | METEOR | BERTScore | NLI | BLEURT |
|---|---|---|---|---|---|
| baseline | 47.57 | 0.380 | 0.916 | 0.852 | 0.176 |
| 1. critic (base) | 47.75 | **0.387** | 0.918 | 0.886 | 0.202 |
| 2. critic (base with full sentences) | 46.06 | 0.376 | 0.917 | **0.898** | **0.212** |
| 3. critic (vanilla LM) | 46.56 | 0.379 | 0.913 | 0.881 | 0.161 |
| 4. critic (fine-tuned LM) | **49.04** | 0.385 | **0.919** | 0.866 | 0.196 |
| 5. critic (fine-tuned LM with full sentences) | 43.74 | 0.372 | 0.909 | 0.861 | 0.123 |

Table 7: Results of automatic evaluation on the WebNLG test set while using beam search (beam size equal to 5).

5. fine-tuned LM – the incorrect next words are sampled from the five most probable tokens according to data-conditioned LM

"The A-Rosa Luna is 125.8m", "The A-Rosa Luna is supplied", "The A-Rosa Luna is powered with", ...