# OpenReview forum: "Critic-Driven Decoding for Mitigating Hallucinations in Data-to-text Generation"
_EMNLP/2023/Conference — EMNLP 2023 Main_

### Official Review · Reviewer_eK7W · 2023-07-29

**Typos Grammar Style And Presentation Improvements:** Eq 3 needs to be better aligned.
**Soundness:** 4

**Excitement:**

3: Ambivalent: It has merits (e.g., it reports state-of-the-art results, the idea is nice), but there are key weaknesses (e.g., it describes incremental work), and it can significantly benefit from another round of revision. However, I won't object to accepting it if my co-reviewers champion it.

**Paper Topic And Main Contributions:**

The authors present a new method to mitigate hallucinations in data-to-text generation.
The method can be combined into an existing system with minimal effort (relative to prior methods) without the need to annotate additional data.

**Reasons To Accept:**

1) The paper is well-written and easy to follow.
2) The idea is simple (in a good way, IMO) and well-motivated.
3) Unlike previous methods, the additional effort for mitigating hallucinations seems to be minimized here, which will likely lead to a broader use for real-world applications.
4) The authors will share their code, which is much appreciated.

**Reasons To Reject:**

1) I know other baselines require additional efforts and might not be directly comparable to your method, but the lack of baselines makes it hard to establish your contribution, as most improvements seem to be marginal (are they statistically significant?)

2) I know it's a very easy critique to make, but I think it's justified in this case. Another dataset can really help in making your findings more robust.

3) Short papers are very challenging to write, but I think this is one of the first papers I read without a "conclusion" section, which harms the paper's self-containment.

**Reproducibility:**

5: Could easily reproduce the results.

**Reviewer Confidence:**

4: Quite sure. I tried to check the important points carefully. It's unlikely, though conceivable, that I missed something that should affect my ratings.

---

> ### Author Rebuttal · Authors · 2023-08-29
>
> Thank you for your comments and for taking the time to read this paper.
>
> >I know other baselines require additional efforts and might not be directly comparable to your method, but the lack of baselines makes it hard to establish your contribution, as most improvements seem to be marginal (are they statistically significant?)
>
> (1) We performed additional statistical tests to compare the results of our baseline and our approach with critic (base with full sentences). As expected, the differences on text quality measures (BLEU, METEOR, BERTScore) are not statistically significant, but the differences on measures targeting hallucinations are statistically significant at the standard alpha=0.05 significance level. For the NLI metric, p-value was equal to 0.0225 (exact test for proportions) and for BLEURT p-value is 0.0360 (unpaired t-test).
>
> >I know it's a very easy critique to make, but I think it's justified in this case. Another dataset can really help in making your findings more robust.
>
> (2) We have run additional experiments on the OpenDialKG dataset with our “critic (base)” approach -- please see our response to reviewer QRL1, item (4)
>
>
> >Short papers are very challenging to write, but I think this is one of the first papers I read without a "conclusion" section, which harms the paper's self-containment.
>
> (3) Thank you for this comment. We will include a formal conclusion section in the revised version of the paper, using the additional page. Currently, the introduction section contains the main highlights of the paper and summarizes the most important findings.

---

### Official Review · Reviewer_GpK3 · 2023-08-04

**Soundness:** 4

**Excitement:**

4: Strong: This paper deepens the understanding of some phenomenon or lowers the barriers to an existing research direction.

**Paper Topic And Main Contributions:**

The paper addresses the problem of on-the-fly mitigation hallucinations in data-to-text generation. Instead of modifying the architecture or generating several hypotheses, the authors propose using a text critic classifier to guide the generation process. This critic is constructed to encourage the generated text to match the input data. The results suggest that the critic approach does not have significant impacts on the quality of the generated text but can mitigate the impact of hallucinations.

**Questions For The Authors:**

A: Your results suggest that your method is able to reduce the overall hallucination rate. However, this reduction does not necessarily translate to higher output quality. Have you analysed what may be the reason for this? Is it that the number of hallucinations is small to impact corpus-level aggregated metrics? Or is it, perhaps of more significance, that some good quality outputs have seen their quality go down after employing the critic?

B: You mention that your critic preserves many outputs (L240) with percentages of around 30-70%. This leads you to mention that the critics "make small changes and only where necessary". However, preserving the outputs does not necessarily mean that the critic only provides changes *only where necessary*. In fact, without further evidence as to how the critic behaves for originally good quality outputs, it is hard to back up this statement. Have you done further experiments that suggest it is indeed the case that the critic only makes changes where necessary?

C: You mention in L274-279 that some methods are more effective than others for reducing hallucinations. It would be very interesting to analyse why this is the case. Have you done some qualitative experiments that provide some hints as to why this may be the case?

**Reasons To Accept:**

- the idea of employing a critic to guide generation is, to the best of my knowledge, novel and has some interesting properties (e.g., not requiring changes to the original LM architecture)
- different ways of training the critic were put forward and analysed extensively, including with human evaluation; the results show that the method holds some promise

**Reasons To Reject:**

- the authors do not compare their proposed methods with any other baselines for mitigation of hallucinations; although they have mentioned these alternatives (e.g., changes to the underlying architecture or generation of multiple hypotheses), they have not shown how these previous methods fare against theirs; this is particularly critical for their story: while this new method does not require changing the underlying architecture or generation of multiple hypotheses, it does require training a classifier that, I would assume, needs to be trained on in-domain data. Moreover, using the critic at test time also incurs inference costs. Thus, this method also comes with its own set of overheads/computation burdens. I believe it is critical to compare with one such method (e.g., generation of multiple outputs w/o critic) both at the level of the output quality, output hallucination and computational cost.
- the analysis of the mitigation method could benefit from finer-grained studies apart from corpus-level/aggregated metrics of quality or measures of edit percentage (e.g., studying how the method impacts the outputs depending on the quality of the original output)
- the authors only tested on one dataset, which may limit the generalisability of their conclusions

**Reproducibility:**

5: Could easily reproduce the results.

**Reviewer Confidence:**

4: Quite sure. I tried to check the important points carefully. It's unlikely, though conceivable, that I missed something that should affect my ratings.

---

> ### Author Rebuttal · Authors · 2023-08-29
>
> Thank you for your comments and for taking the time to read this paper.
>
> >the authors do not compare their proposed methods with any other baselines for mitigation of hallucinations; although they have mentioned these alternatives (e.g., changes to the underlying architecture or generation of multiple hypotheses), they have not shown how these previous methods fare against theirs; this is particularly critical for their story: while this new method does not require changing the underlying architecture or generation of multiple hypotheses, it does require training a classifier that, I would assume, needs to be trained on in-domain data. Moreover, using the critic at test time also incurs inference costs. Thus, this method also comes with its own set of overheads/computation burdens. I believe it is critical to compare with one such method (e.g., generation of multiple outputs w/o critic) both at the level of the output quality, output hallucination and computational cost.
>
> (1) Thank you for this comment. One simple method which generates multiple outputs and generally tends to offer texts of higher quality is beam search. We run additional experiments with beam size equal to 5 and present the result in the table below. The improvements for this stronger baseline are consistent with these reported in the paper for greedy decoding.
> |                                            | BLEU       | METEOR    | BERTScore | NLI       | BLEURT    |
> |--------------------------------------------|------------|-----------|-----------|-----------|-----------|
> | baseline                                   | 47,573     | 0,380     | 0,916     | 0,852     | 0,176     |
> | critic (base)                              | 47,752     | **0,387** | 0,918     | 0,886     | 0,202     |
> | critic (base with full sentences)          | 46,056     | 0,376     | 0,917     | **0,898** | **0,212** |
> | critic (vanilla LM)                        | 46,561     | 0,379     | 0,913     | 0,881     | 0,161     |
> | critic (fine-tuned LM)                     | **49,043** | 0,385     | **0,919** | 0,866     | 0,196     |
> | critic (fine-tuned LM with full sentences) | 43,740     | 0,372     | 0,909     | 0,861     | 0,123     |
>
> >the analysis of the mitigation method could benefit from finer-grained studies apart from corpus-level/aggregated metrics of quality or measures of edit percentage (e.g., studying how the method impacts the outputs depending on the quality of the original output)
>
> (2)  Thank you for this comment. We’ve already provided some additional insights, such as the number of added/removed words and modified outputs reported in the paper. We will report some additional fine-grained results on the additional page provided upon acceptance.
>
> >the authors only tested on one dataset, which may limit the generalisability of their conclusions
>
> (3) We have run additional experiments on the OpenDialKG dataset with our “critic (base)” approach -- please see our response to reviewer QRL1, item (4)
>
> >A: Your results suggest that your method is able to reduce the overall hallucination rate. However, this reduction does not necessarily translate to higher output quality. Have you analysed what may be the reason for this? Is it that the number of hallucinations is small to impact corpus-level aggregated metrics? Or is it, perhaps of more significance, that some good quality outputs have seen their quality go down after employing the critic?
>
> (4) Thank you for this comment - we were also interested in this matter.
>
> First of all, we would like to point out that our method does improve the text quality metrics in a vast majority of configurations, albeit slightly.
>
> We attribute the lack of a larger metric gain to two factors. (1) Our method generally modifies a small number of words in the generated texts, which can be hard to capture by n-gram based metrics. (2) The automatic measures are generally imperfect and have some shortcomings. The manual evaluation of text quality (expressed by ranking the model outputs) demonstrated that all critic approaches obtained better average rank than the baseline.
>
> >B: You mention that your critic preserves many outputs (L240) with percentages of around 30-70%. This leads you to mention that the critics "make small changes and only where necessary". However, preserving the outputs does not necessarily mean that the critic only provides changes only where necessary. In fact, without further evidence as to how the critic behaves for originally good quality outputs, it is hard to back up this statement. Have you done further experiments that suggest it is indeed the case that the critic only makes changes where necessary?
>
> (5) It is true that the critic can sometimes suggest changes which are not correct, but this is relatively rare. For instance, critic (base) introduces errors to ~5% of originally correctly generated sentences while correcting >32% of incorrect outputs (in terms of hallucinations).
>
> >C: You mention in L274-279 that some methods are more effective than others for reducing hallucinations. It would be very interesting to analyse why this is the case. Have you done some qualitative experiments that provide some hints as to why this may be the case?
>
> (6) We suspect that lower improvements of critics trained on data generated with fine-tuned LMs are related to the fact that LMs often generate correct data verbalizations that differ from the reference. In such a situation, the text suggested by LM is correct but the comparison to the reference constructs a negative example for the critic's training data. This can lead the critic both to over-focus on generating the exact reference and to learn to negatively assess other possible correct data verbalizations.

---

### Official Review · Reviewer_QRL1 · 2023-08-06

**Soundness:** 4

**Excitement:**

2: Mediocre: This paper makes marginal contributions (vs non-contemporaneous work), so I would rather not see it in the conference.

**Missing References:**

- One reason that why this can show improvement is that the critic classifier is trained to learn those unlikelihood samples (e.g., NEURAL TEXT DEGENERATION WITH UNLIKELIHOOD TRAINING). Even though this line of work focuses more on the training stage not inference stage, but I'd recommend the authors to check and compare.

**Paper Topic And Main Contributions:**

* This paper is about a decoding strategy that takes a classifier output into account for resampling.
* The main contribution is the exploration of showing different results on negative sample generation, and showed that the in-domain generation results can reduce hallucinations.

**Questions For The Authors:**

- What is the agreement score from 5 annotators?

**Reasons To Accept:**

- It showed several negative sampling strategies and its performance on guiding a trained generator, and showed that by human evaluation it can reduce hallucination.

**Reasons To Reject:**

- It only showed on one dataset and evaluated “in-domain” results. For hallucination, it is more important to show its generalization ability, especially nowadays we have generators that are pretrained on billions of tokens already.

- The proposed critic sampling strategy is not novel. This paper reminds me of PPLM (Plug and Play Language Models: A Simple Approach to Controlled Text Generation) and its literature of controllable generation. To me this is a subset of the controllability and focusing on reducing hallucination.

- More hallucination tasks are expected to be evaluated, especially summarization tasks and open-domain QA tasks.

- More hallucination reduction methods are expected to be compared. For examples, Improving Factual Consistency in Summarization with Compression-Based Post-Editing, or check section 5 in this paper to get some ideas (Survey of Hallucination in Natural Language Generation).

**Reproducibility:**

4: Could mostly reproduce the results, but there may be some variation because of sample variance or minor variations in their interpretation of the protocol or method.

**Reviewer Confidence:**

3: Pretty sure, but there's a chance I missed something. Although I have a good feel for this area in general, I did not carefully check the paper's details, e.g., the math, experimental design, or novelty.

---

> ### Author Rebuttal · Authors · 2023-08-29
>
> Thank you for your comments and for taking the time to read this paper.
>
> >It only showed on one dataset and evaluated “in-domain” results. (...)
>
> (1) Our approach is evaluated on both in-domain and out-of-domain data. The test data of the WebNLG dataset contains about 46% of instances from categories not present in the training data. We add the fine-grained results below. The in-domain results are naturally higher, but one can observe consistent improvements on both in-domain and out-of-domain data. Additionally, we also offer results on one more dataset in (4).
>
> |                                     |   NLI   |               |           |  BLEURT |               |           |
> |-------------------------------------|:-------:|:-------------:|:---------:|:-------:|:-------------:|:---------:|
> |                                     | overall | out-of-domain | in-domain | overall | out-of-domain | in-domain |
> | baseline                            | 0,841   | 0,783         | 0,889     | 0,128   | -0,026        | 0,257     |
> | critic (base)                       | 0,855   | 0,801         | 0,901     | **0,155**   | **0,010**         | **0,277**     |
> | critic (base with full sentences)   | **0,868**   | **0,820**         | **0,909**     | 0,153   | 0,007         | 0,274     |
> | critic (vanilla LM)                 | 0,859   | 0,811         | 0,900     | 0,139   | -0,002        | 0,258     |
> | critic (fine-tuned LM)              | 0,834   | 0,772         | 0,886     | 0,128   | -0,021        | 0,254     |
> | critic (fine-tuned LM w/full sent.) | 0,839   | 0,779         | 0,889     | 0,136   | -0,013        | 0,261     |
>
>
> >The proposed critic sampling strategy is not novel. This paper reminds me of PPLM (Plug and Play Language Models: A Simple Approach to Controlled Text Generation) and its literature of controllable generation. To me this is a subset of the controllability and focusing on reducing hallucination.
>
> (2) We agree that our approach is related to PPLM (hence Dathathri et al. are cited in our Related Work section). However, we believe that there are important differences:
>
> - PPLM performs “ex post facto” optimization of the LM activation values and requires gradient computations during decoding. In contrast, our critic-based approach only modifies the probabilities of the next token. Even though PPLM does not require LM retraining, it does require access to its implementation to cache historical activation values of neurons and to perform specific backpropagation during decoding. Our approach can potentially work even with non-differentiable models (since only output probabilities are modified) and does not require any changes in the LM implementation. Note that in practice it may be difficult to access some (e.g. proprietary) model implementations.
>
> - Second, the PPLM attribute model is conditioned on specific historical activation values of a given LM (see Eq. 3 of Dathathri et al.). In contrast, some of our critics (e.g. the well-performing critic-base) have been trained completely independently of the LM itself and, once trained, can be straightforwardly used in decoding any LM.
>
> - Although controllability and hallucination mitigation are related, we argue that controlling sentiment or term usage (Dathathri et al.) is easier than ensuring the fidelity of generated text (our paper).
>
> >More hallucination tasks are expected to be evaluated, especially summarization tasks and open-domain QA tasks.
>
> (3) We agree that more tasks could be evaluated, but the scope of the current short paper is limited to the challenging data-to-text task.
>
> >More hallucination reduction methods are expected to be compared. For examples, Improving Factual Consistency in Summarization with Compression-Based Post-Editing, or check section 5 in this paper to get some ideas (Survey of Hallucination in Natural Language Generation).
>
> (4) Thank you for pointing out interesting works. Unfortunately, the paper you mentioned and also other methods from Section 5.2.3 (Post-Processing methods, which are the most related to our work) are designed for the summarization task and are not directly applicable to our data-to-text task. This shows that there is a certain research gap that our paper fits in.
> The only work on a related task is Neural Path Hunter (NPH, by Dziri et al.), which post-edits utterances from a dialog agent using small knowledge graphs. We were not able to replicate their experiments in this short time due to problems with unresolved dependencies etc., but  we will contact Dziri et al. to compare both methods directly and provide the results in the final version. Currently, we were able to run our approach with “critic (base)” on their OpenDialKG dataset, using the data in the same way as WebNLG (i.e. without taking dialog history into account). On measures targeting hallucinations, we obtained +5% on NLI and +3% on BLEURT. Basic word-overlap metrics are lower, but our qualitative assessment did not confirm any quality drop. For the final version of the paper, we'll prepare the results for all the critics, including a variant conditioned on dialog history.
>
> |                | BLEU  | METOR | BERTScore | NLI   | BLEURT |
> |----------------|-------|-------|-----------|-------|--------|
> | baseline       | **11,73** | **0,149** | **0,775**     | 0,748 | -0,933 |
> | critic (base)  | 9,66  | 0,137 | 0,771     | **0,796** | **-0,905** |
>
> >What is the agreement score from 5 annotators?
>
> (5) In our study, we didn’t have overlapping annotations so the inter-annotation agreement was not originally measured. We asked one annotator to re-annotate 10 (x 6 model outputs) instances originally annotated by a different annotator. 86% of annotations were identical. In terms of Cohen’s kappa, we obtained 0.19 agreement for minor hallucinations, 0.68 for major, 0.88 for omissions, 0.48 for repetitions and 0.07 for disfluencies.
>
> >One reason that why this can show improvement is that the critic classifier is trained to learn those unlikelihood samples (e.g., NEURAL TEXT DEGENERATION WITH UNLIKELIHOOD TRAINING). Even though this line of work focuses more on the training stage not inference stage, but I'd recommend the authors to check and compare.
>
> (6) Thank you for pointing out this interesting work. We will include it in the revised version of Related Work section.

---

### Meta-Review · Area_Chair_A2WP · 2023-09-19

**Recommendation:** 4

**Metareview:**

This paper presents a method for reducing hallucinations by additionally incorporating a classifier's probability to assess the correctness between the generated text so far during decoding. Reviewers agree that this method is useful, simple (in a positive way), and easy to apply. The idea is also well-motivated, and the authors have shown extensive results.

A common drawback of this work, as raised by reviewers, is that the paper evaluates its method only on a single dataset, thereby raising questions about its generality. Additionally, there is a lack of comparison with other hallucination mitigation baselines.

---

### Decision · Program_Chairs · 2023-10-07

**Decision:**

Accept-Main

**Comment:**

This paper presents a method for reducing hallucinations by additionally incorporating a classifier's probability to assess the correctness between the generated text so far during decoding. Reviewers agree that this method is useful, simple (in a positive way), and easy to apply. The idea is also well-motivated, and the authors have shown extensive results.

A common drawback of this work, as raised by reviewers, is that the paper evaluates its method only on a single dataset, thereby raising questions about its generality. Additionally, there is a lack of comparison with other hallucination mitigation baselines.